# LST-BENCH: A BENCHMARK FOR LONG SEQUENCE TIME-SERIES FORECASTING TASK

## ABSTRACT

This paper introduces LST-Bench, a comprehensive benchmark designed for evaluating long sequence time-series forecasting(LSTF) models. This benchmark has been developed in response to recent advancements in deep learning methods in the field of LSTF tasks. LST-Bench includes Transformer-based, MLP-based, CNN-based, and RNN-based models, evaluating the performance of 11 major forecasting models on a set of commonly used 7 datasets and 7 new datasets that we have introduced. We conduct a thorough analysis of the experimental results, including the overall prediction performance of models and their generalization across different prediction lengths and datasets. Notably, we found that regardless of the model architecture, the phenomenon referred to as "Degeneracy" occurs when the model's predictions consistently maintain a low Mean Squared Error value but are characterized by repetitive and simplistic pattern generation, thus losing the meaningfulness of the predictions. Also, the model's optimal performance is very close to its performance after training for just one epoch. These two phenomenons emphasize the need for further investigation. Our LST-Bench will serve as a valuable resource for advancing research in the field of time series forecasting.

## 1 INTRODUCTION

Time series analysis plays a crucial role in practical applications, and its significance is evident in its capability to provide strong support for decision-making, trend forecasting, and problem-solving by mining patterns and trends in historical data, especially for complex problems that change over time. Time series forecasting has always been a central issue in the field of time series. In 2021, Zhou et al. (2021) introduced the concept of long-term time series forecasting for the first time, leading to the emergence of many new models in this field, including variations of Transformer (Vaswani et al. (2017)), CNN, RNN, and MLP/FC structures. In 2023, Zeng et al. (2023) raised an important question about the practical effectiveness of the Transformer structure in long sequence time-series forecasting. At the same time, Nie et al. (2022) addressed the concerns about the effectiveness of Transformer-based solutions for LSTF problem in their research and successfully achieved performance surpassing by a simple MLP-based model on widely used datasets, thereby providing valuable insights and innovations for the development of the field of time series forecasting.

To comprehensively evaluate the currently widely used long sequence time-series forecasting models, we propose the LST (Long Sequence Time Series) benchmark, which conducts a comprehensive assessment of the major 11 time series forecasting models. In addition to covering common datasets such as Weather, Traffic, Electricity, and 4 ETT datasets, we introduce 7 new datasets called **NEW**, enriching the data samples and diversity for time series evaluation, providing a broader range of experimental resources and challenges for the time series field. Simultaneously, we conducted experiments at four different prediction lengths, covering prediction intervals from 96 to 720 time steps, to evaluate the performance differences of the models in short-term and long-term forecasting. Furthermore, we conducted experiments related to the length of the model input sequence, prediction consistency, execution speed, and generalization, to compare the comprehensive performance of different models and elaborate on their similarities and differences.

After benchmark testing experiments, we propose some new academic viewpoints and questions for the current time series forecasting tasks. Firstly, we notice that in the context of addressing

LSTF problems, all models exhibit significantly faster convergence compared to models within the NLPDevlin et al. (2018), CVLiu et al. (2021b), and speech recognizationDong et al. (2018) domains, which indicate that it does not fully capture the complex patterns and information in the time-series data. Secondly, in nearly all models, we have observed a phenomenon we named as **Degeneracy**, which signifies that the model's predictions exhibit lower MSE/MAE values, yet in fact, the forecasted outcomes tend to manifest as periodic repetitions, failing to genuinely predict the future data. We believe it is because the current metrics fail to adequately reflect the actual requirements of time series forecasting tasks or are influenced by certain data characteristics. Addressing this issue requires a reevaluation or redefinition of evaluation metrics applicable to time series forecasting to more accurately measure model performance, model design, and improvement on that. These issues require in-depth research and discussion to further advance the field.

## 2 RELATED WORK

### 2.1 TIME SERIES PREDICTION MODEL

#### 2.1.1 TRANSFORMER BASED

Transformer architecture stands out as one of the most prominent and widely applied model architectures in the field of deep learning. After making remarkable strides in NLP tasks, it swiftly found its way into the domain of time series analysis, giving rise to a continuous stream of innovative models built upon this foundational structure. In Wu et al. (2020), a Transformer structure similar to GPT was first used to tackle time series forecasting tasks with promising results.

While demonstrating the effectiveness of the Transformer architecture in the field of time series, it still encounters challenges related to quadratic computational complexity and high memory usage, particularly when dealing with long sequence forecasting. To address these problems, LogTrans Li et al. (2019) combined CNN and Transformer and introduced a convolutional self-attention mechanism. In the attention layer, causal convolutions are used to generate queries and keys. Additionally, it introduced a log-sparse mask, reducing the model's complexity from $\mathcal{O}(L^2)$ to $\mathcal{O}(L \log L)$. Informer Zhou et al. (2021) observed that attention scores in the original Transformer follow a long-tail distribution. Therefore, it selects important queries for computation based on query-key similarity, achieving similar computational complexity as LogTrans. Informer also designs a generative-style decoder for direct long-term forecasting to avoid cumulative errors in one-step predictions for long-term forecasts. Pyraformer Liu et al. (2021a) structures multiple layers into a pyramid-like structure, reducing the number of similarity calculations between nodes, thus lowering complexity while ensuring the effectiveness of capturing long-term dependencies.

On the other hand, some research explores frequency-domain self-attention mechanismsTorrence & Compo (1998) and decomposition mechanismsCleveland et al. (1990) in time series modeling. Autoformer Wu et al. (2021) introduces a short-term trend decomposition architecture and employs autocorrelation mechanisms in its attention module. Unlike previous attention mechanisms, autocorrelation measures the time-delay similarity between input signals and aggregates the top-k similar subsequences to produce outputs with $\mathcal{O}(L \log L)$ complexity. FEDformer Zhou et al. (2022) calculates attention mechanisms in the frequency domain using Fourier and wavelet transforms, randomly selecting a fixed-size subset of frequencies, achieving linear complexity.

PatchTSTNie et al. (2022) introduced the concept of patching for the first time in the field of time series, thus disrupting the traditional approach of predominantly using point-wise attention in the most of existing models. The introduction of patches enhances the model's ability to learn information from data sub-series, with a greater focus on the interrelatedness between these sub-series. The use of sub-series also facilitates the model in capturing data periodicity more effectively, as it readily identifies repetitions or similarities among sub-series while attending to their relationships. In the experimental section, we observed that models employing sub-series attention mechanisms and decomposition outperform those using point-wise attention.

#### 2.1.2 MLP BASED

The N-BEATS Oreshkin et al. (2019) model is a novel time series forecasting backbone introduced in 2019, which achieves time series forecasting solely through fully connected layers. In N-BEATS,

the input is limited to a single time series and does not allow for the inclusion of external features such as date information, holiday information, or attribute information, which are important in time series forecasting tasks. To address this limitation, N-BEATSx Olivares et al. (2023) was proposed, allowing the integration of external features into the N-BEATS model. DLinear Zeng et al. (2023) questions the necessity of using Transformers for LSTF problem, and shows that a simpler MLP-based model can achieve better results compared to some Transformer baselines through empirical studies.

### 2.1.3 RNN BASED

LSTM (Long Short-Term Memory) Chung et al. (2014) was introduced as an improvement over basic RNNs to address the vanishing gradient or exploding gradient problem when dealing with long-term dependencies. LSTM uses gate mechanisms, including the forget gate, input gate, and output gate, to control the flow of information. This allows LSTM to capture long-term dependencies and is suitable for tasks that require memorizing information over long intervals. GRU (Gated Recurrent Unit) Cho et al. (2014) is a structure similar to LSTM but simplifies the gate mechanisms to only two: the update gate and the reset gate. GRU performs comparably to LSTM in some cases but has fewer parameters and trains faster.

### 2.1.4 CNN BASED

CNN (Convolutional Neural Network) is typically composed of layers such as the input layer, convolutional layers, activation layers, pooling layers, and fully connected layers. It is primarily used for tasks like image classification, but in recent years, CNN-based variants have been applied to time series forecasting tasks as well. SCINet Liu et al. (2022) constructs basic blocks called SCI-blocks. These blocks sub-sample the input data into two sub-sequences and use different convolutional filters to extract features from each sub-sequence, preserving information about different features. To mitigate the impact of information loss during subsampling, SCINet introduces learning of convolutional features between sequences within each SCI-Block.

## 2.2 EXISTING BENCHMARK

At present, there is a notable scarcity of benchmarks for time series forecasting, and the few benchmarks available are considerably dated, with limited representation of deep learning methodologies. Libra Bauer et al. (2021) only includes statistical and machine learning methods such as ARIMAZhang (2003), XgboostChen & Guestrin (2016), and Random ForestBreiman (2001). Ismail et al. (2020) focuses specifically on addressing the interpretabilityHooker et al. (2019) of time series prediction problems. Our work can be considered as the first benchmark for this problem after the widespread application of deep learning in time series forecasting.

## 3 EXPERIMENT DESIGN

### 3.1 PROBLEM SETUP

The definition of a time series forecasting problem is as follows. Given the input sequences $\mathcal{X}_t = \{\mathbf{x}_1^t, ..., \mathbf{x}_{L_x}^t | \mathbf{x}_i^t \in \mathbb{R}^{d_x}\}$ at time $t$, time scries forecasting is to predict the output sequences $\mathcal{Y}_t = \{\mathbf{y}_1^t, ..., \mathbf{y}_{L_y}^t | \mathbf{y}_i^t \in \mathbb{R}^{d_y}\}$, where $\mathbf{x}_i^t$(or $\mathbf{y}_i^t$) is a subserie with dimension $d_x$(or $d_y$) at the t-th moment. In the long sequence time-series forecasting (LSTF) problem, the following more conditions are required. Firstly, the output's length $\mathcal{Y}_t$ is longer than previous work (Cho et al. (2014); Sutskever et al. (2014)), like predicting 48 points or less(Hochreiter & Schmidhuber (1997);Li et al. (2017);Yu et al. (2017);Liu et al. (2020);Qin et al. (2017);Wen et al. (2017)); Secondly, the feature dimension of the output is not limited to univariate case($d_y \leq 1$).

## 3.2 DATASETS

### 3.2.1 DATASETS DESCRIPTION

Our dataset **NEW** consists of two years' worth of 15-minute-level data from the power industry, primarily used for equipment monitoring in the electricity sector. The data will be made open source.

We perform experiments on 7 commonly used public datasets, including 4 **ETT** datasets Zhou et al. (2021), **Weather**, **ECL**, **Traffic**, and our 7 **NEW** datasets. The train/val/test is 12/4/4 months for ETT and NEW. As for **ETT**, **Weather**, **ECL**, **Traffic**, the ratio of train/val/test is 7:2:1.The 7 commonly used public datasets' details please refer to AppendixA.6.

## 3.3 FAIRNESS

In order to ensure fairness, we restricted several key conditions to the same:

**Platform:** All models were trained and tested on the single Nvidia V100 32GB GPU.

**Metrics:** We chose two metrics: Mean Square Error(MSE) and Mean Absolute Error(MAE), where $MSE = \frac{1}{n}\sum_{i=1}^{n}(y - \hat{y})^2$ and $MAE = \frac{1}{n}\sum_{i=1}^{n}|y - \hat{y}|$. For each model, we test on each sliding prediction window and roll the whole test set with stride = 1. The final result is the average of the results of all Windows.

**Dataset:** All models were trained and tested using the data and partitioning methods mentioned in Section 3.2. The input data length of all experiments is unified to 336, and the predicted data length is unified to 192.

**Implemention:** Unless otherwise specified, for all models evaluated in the benchmark, we used the model code and scripts from the model authors' open-source code repository. Hyperparameters and scripts were kept consistent with the defaults. We do not implement RevIN(Kim et al. (2021)), which is a general normalization method. For specific model replication details, please refer to Appendix A.2.

## 4 RESULTS AND EVALUATION

**Overall Prediction Accuracy**: Table 1 summarizes the multivariate time series prediction results of the top 6 models in terms of overall performance on the 7 **NEW** datasets we provided. For full benchmark results, please refer to Appendix A.1. The experiment is designed to test the models' overall prediction performance and their ability to handle longer prediction lengths.

We find that the performance rankings of the models (see Table 2) can be broadly categorized into three tiers and others. The performance of models in each tier is closely matched:

- The **first-tier** models are **PatchTST** and **DLinear**, with average rankings 1 to 2.
- The **second-tier** models are **SCINet** and **FEDformer**, with average rankings 3 to 4.
- The **third-tier** models are **Autoformer** and **N-BEATS**, with average rankings 5 to 6.

From a model architecture perspective, the high-performing models are those based on Transformer architecture, MLP/FC architecture, and CNN architecture, whereas RNN architecture models consistently underperform.

From figure 1, the rankings of various models are very stable across different datasets and prediction lengths. Upon averaging the results across all datasets, minimal fluctuations in model rankings are evident, with most models experiencing marginal variations of approximately one position. From more details about the stability of the rankings, please refer to Appendix A.5.

**Results of prediction consistency within different prediction length:** We assume that the prediction results of a model should be more and more inaccurate with the prediction length becomes longer. We proceeded to verify the consistency of model predictive accuracy with changing prediction horizons. From Table 3, within the prediction lengths of $\{336, 720\}$, all Transformer structure models exhibit some degree of inconsistency on several datasets. Conversely, MLP/FC structure

Table 1: Multivariate long-term series forecasting results on 7 NEW datasets with input length $I = 336$ and prediction length $O \in \{96, 192, 336, 720\}$. A lower MSE indicates better performance, and the best results are highlighted in bold. For full benchmark results please refer to Appendix A.1.

| Prediction | | 96 | | 192 | | 336 | | 720 | |
|---|---|---|---|---|---|---|---|---|---|
| | | MSE | MAE | MSE | MAE | MSE | MAE | MSE | MAE |
| NEW1 | PatchTST | **0.079** | 0.163 | 0.141 | 0.251 | 0.161 | 0.261 | 0.318 | 0.425 |
| | DLinear | **0.079** | **0.161** | **0.111** | **0.200** | **0.146** | **0.237** | 0.247 | 0.344 |
| | SCINet | 0.088 | 0.169 | 0.128 | 0.213 | 0.167 | 0.252 | 0.242 | **0.312** |
| | FEDformer | 0.105 | 0.220 | 0.136 | 0.253 | 0.180 | 0.285 | **0.228** | 0.317 |
| | N-BEATS | 0.108 | 0.211 | 0.226 | 0.324 | 0.278 | 0.361 | 0.339 | 0.398 |
| | Autoformer | 0.252 | 0.349 | 0.264 | 0.362 | 0.355 | 0.420 | 0.380 | 0.427 |
| NEW2 | PatchTST | 0.105 | 0.214 | 0.156 | 0.267 | 0.183 | 0.280 | 0.330 | **0.377** |
| | DLinear | **0.079** | **0.174** | **0.110** | **0.209** | **0.148** | **0.245** | 0.320 | 0.389 |
| | SCINet | 0.089 | 0.189 | 0.133 | 0.239 | 0.178 | 0.279 | 0.312 | 0.378 |
| | FEDformer | 0.143 | 0.274 | 0.183 | 0.308 | 0.233 | 0.343 | **0.304** | 0.387 |
| | N-BEATS | 0.118 | 0.234 | 0.231 | 0.341 | 0.292 | 0.382 | 0.390 | 0.432 |
| | Autoformer | 0.300 | 0.383 | 0.279 | 0.363 | 0.362 | 0.423 | 0.474 | 0.472 |
| NEW3 | PatchTST | 0.210 | 0.321 | 0.269 | 0.372 | **0.312** | **0.402** | **0.400** | **0.463** |
| | DLinear | **0.200** | **0.311** | **0.265** | **0.366** | 0.323 | 0.412 | 0.455 | 0.502 |
| | SCINet | 0.233 | 0.344 | 0.361 | 0.436 | 0.554 | 0.545 | 0.825 | 0.676 |
| | FEDformer | 0.304 | 0.409 | 0.346 | 0.435 | 0.383 | 0.461 | 0.465 | 0.510 |
| | N-BEATS | 0.255 | 0.359 | 0.321 | 0.412 | 0.361 | 0.437 | 0.431 | 0.485 |
| | Autoformer | 0.379 | 0.460 | 0.451 | 0.506 | 0.478 | 0.507 | 0.483 | 0.509 |
| NEW4 | PatchTST | 0.506 | 0.404 | 0.530 | 0.435 | **0.621** | **0.478** | **0.844** | **0.571** |
| | DLinear | **0.397** | **0.373** | **0.493** | **0.422** | 0.626 | 0.487 | 0.905 | 0.616 |
| | SCINet | 0.514 | 0.415 | 0.636 | 0.476 | 0.785 | 0.545 | 1.072 | 0.677 |
| | FEDformer | 0.543 | 0.458 | 0.628 | 0.495 | 0.713 | 0.523 | 0.928 | 0.605 |
| | N-BEATS | 0.625 | 0.466 | 0.803 | 0.561 | 0.998 | 0.631 | 1.202 | 0.710 |
| | Autoformer | 0.900 | 0.624 | 1.017 | 0.667 | 1.096 | 0.676 | 1.419 | 0.760 |
| NEW5 | PatchTST | 0.191 | 0.296 | 0.240 | 0.342 | 0.297 | 0.385 | 0.409 | 0.461 |
| | DLinear | **0.181** | **0.283** | **0.220** | **0.320** | **0.276** | **0.369** | **0.395** | **0.455** |
| | SCINet | 0.194 | 0.303 | 0.251 | 0.354 | 0.327 | 0.413 | 0.435 | 0.485 |
| | FEDformer | 0.239 | 0.353 | 0.290 | 0.393 | 0.350 | 0.432 | 0.444 | 0.485 |
| | N-BEATS | 0.393 | 0.391 | 0.639 | 0.524 | 0.802 | 0.598 | 0.986 | 0.671 |
| | Autoformer | 0.876 | 0.650 | 0.860 | 0.637 | 1.017 | 0.694 | 1.220 | 0.773 |
| NEW6 | PatchTST | 0.284 | 0.364 | 0.306 | 0.392 | 0.377 | 0.440 | 0.434 | 0.491 |
| | DLinear | 0.263 | 0.356 | **0.294** | **0.385** | **0.332** | **0.419** | 0.432 | 0.497 |
| | SCINet | 0.275 | 0.366 | 0.316 | 0.407 | 0.357 | 0.435 | 0.469 | 0.512 |
| | FEDformer | 0.347 | 0.430 | 0.377 | 0.452 | 0.407 | 0.473 | 0.497 | 0.529 |
| | N-BEATS | **0.207** | **0.322** | 0.296 | 0.396 | 0.347 | 0.437 | **0.414** | **0.484** |
| | Autoformer | 0.405 | 0.488 | 0.413 | 0.488 | 0.380 | 0.468 | 0.427 | 0.490 |
| NEW7 | PatchTST | 0.663 | 0.504 | 0.762 | 0.552 | 0.837 | 0.594 | **0.944** | **0.654** |
| | DLinear | **0.629** | **0.487** | **0.699** | **0.531** | **0.782** | **0.574** | 1.050 | 0.700 |
| | SCINet | 0.654 | 0.527 | 0.744 | 0.566 | 0.861 | 0.619 | 1.099 | 0.720 |
| | FEDformer | 0.979 | 0.643 | 1.094 | 0.667 | 1.241 | 0.733 | 1.123 | 0.734 |
| | N-BEATS | 0.754 | 0.588 | 0.945 | 0.668 | 1.027 | 0.712 | 1.096 | 0.768 |
| | Autoformer | 1.060 | 0.744 | 1.000 | 0.725 | 1.108 | 0.775 | 1.233 | 0.819 |

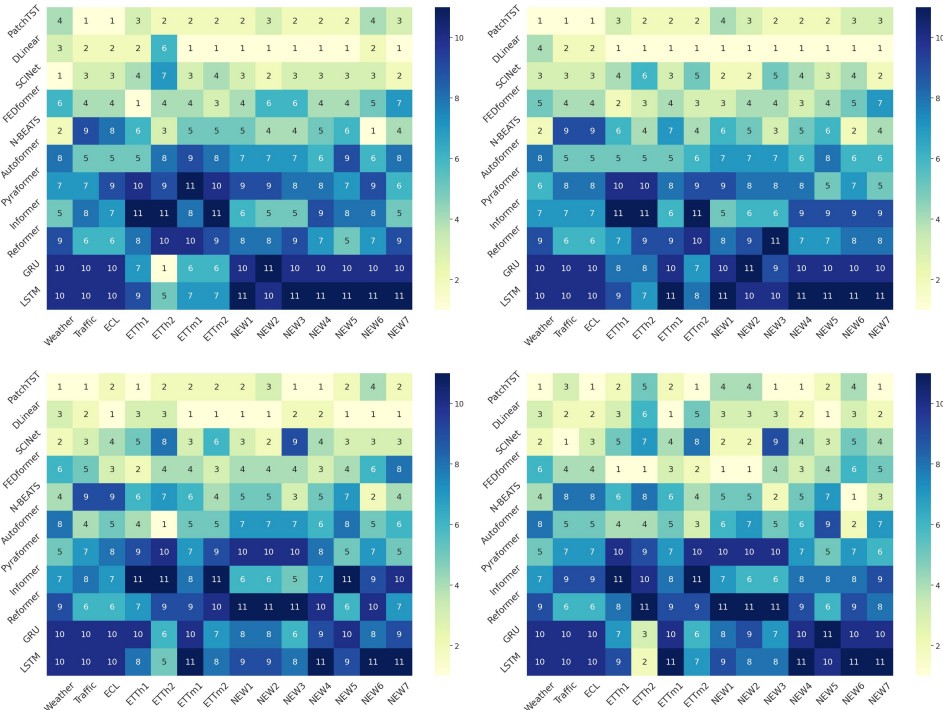

Figure 1: The average ranking of the model across different prediction lengths. From left to right, top to bottom, these are rankings of various models on a specific dataset at prediction lengths in $\{96, 192, 336, 720\}$. The lighter the color, the higher the ranking. The number in the center of each square represents the model's ranking on that dataset. Firstly, we sorted the predictions made by all models for each prediction horizon within each dataset. Then, we averaged the rankings across different prediction horizons within the same dataset. In cases where rankings were tied, they were considered equivalent.

Table 2: The average ranking of models across different prediction lengths. Obtained by taking the average of model performance rankings across all datasets.

|  | PatchTST | Dlinear | SCINet | FEDformer | N-BEATS | Autoformer | Pyraformer | Informer | Reformer | GRU | LSTM |
|---|---|---|---|---|---|---|---|---|---|---|---|
| 96 | 2.4 | 1.8 | 3.1 | 4.4 | 4.8 | 7.0 | 8.5 | 7.6 | 7.9 | 8.6 | 9.6 |
| 192 | 2.2 | 1.4 | 3.5 | 3.9 | 5.1 | 6.1 | 7.8 | 8.1 | 8.1 | 9.5 | 10.0 |
| 336 | 1.9 | 1.6 | 4.1 | 4.3 | 5.4 | 5.6 | 7.9 | 8.4 | 8.7 | 8.6 | 9.3 |
| 720 | 2.3 | 2.8 | 4.2 | 3.2 | 5.1 | 5.4 | 7.9 | 8.4 | 8.8 | 8.6 | 9.1 |
| avg. | 1.7 | 1.8 | 3.7 | 3.8 | 5.2 | 5.8 | 7.6 | 8.1 | 8.4 | 8.9 | 9.7 |

models demonstrate excellent performance, essentially upholding consistency across the entirety of the datasets. The performance of the CNN structural model is second only to the MLP/FC structural model. RNN structure models appear to struggle with consistency, to the extent that they don't even maintain consistency on a single dataset. When the prediction horizon is reduced to 336, it becomes easier for the models to maintain consistency. However, the three poorest-performing models, Autoformer, RNN, and LSTM, show no improvement even when the requirements are relaxed. Regarding the predictive performance drop of the model from 96 to 720, please refer to Appendix A.3.

Table 3: The number of datasets on which the model exist inconsistency across different prediction lengths. Within a certain prediction range, if the model performs better on a dataset with longer prediction lengths, count once.

| Numbers of datasets within 720 | | | | Numbers of datasets within 336 | | | |
|---|---|---|---|---|---|---|---|
| PatchTST | 1/14 | Pyraformer | 3/14 | PatchTST | 1/14 | Pyraformer | 2/14 |
| DLinear | 0/14 | Informer | 7/14 | DLinear | 0/14 | Informer | 4/14 |
| SCINet | 1/14 | Reformer | 3/14 | SCINet | 0/14 | Reformer | 3/14 |
| FEDformer | 2/14 | GRU | 11/11 | FEDformer | 0/14 | GRU | 11/11 |
| N-BEATS | 0/14 | LSTM | 11/11 | N-BEATS | 0/14 | LSTM | 11/11 |
| Autoformer | 9/14 | AvgPred | 2/11 | Autoformer | 9/14 | AvgPred | 2/11 |

**Model generality on different dataset:** For the rankings of models, as shown in the figure 2, we find that there are significant performance differences among models from different tiers, and there is almost no difference in evaluation when using both MSE and MAE. Under these two metrics, Transformer-based models (PatchTST, FEDformer, Autoformer) in the first three tiers have demonstrated better performance on longer prediction length: as the prediction length increases, the model's ranking also improves. Models based on MLP/FC and CNN architectures (DLinear, SCINet, N-BEATS) have an advantage in shorter prediction lengths, but gradually get surpassed by Transformer-based models in their tiers as the prediction length increases. For models outside of the three tiers, the relationship between prediction horizon and prediction performance is unclear due to their inadequate performance.

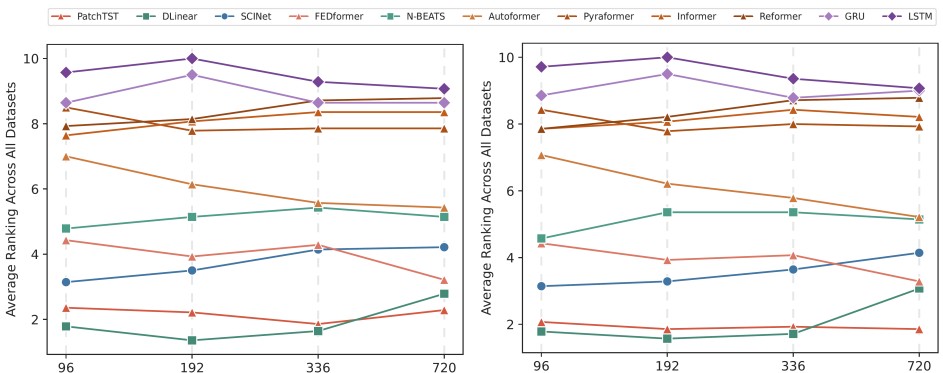

Figure 2: The average ranking of models according to MSE and MAE across all datasets for prediction length in $\{96, 192, 336, 720\}$.

For the MSE of models, as shown in the figure 3. The median of MSE values predicted by the models across all datasets generally follows an increasing trend as the prediction length increases, except for LSTM and GRU. Among the models in the first three tiers, PatchTST and DLinear in tier 1 have relatively concentrated MSE values within the middle 50% range, with very few outliers, indicating strong stability in their performance across different datasets. In tier 2, FEDformer has a smaller interquartile range compared to tier 1 models but has the highest number of outliers, suggesting sensitivity of its performance to the datasets. Similarly, SCINet in tier 2 has a much larger interquartile range than FEDformer but lacks outliers. Autoformer exhibits characteristics similar to FEDformer. RNN-based models consistently exhibit a uniform MSE distribution across all prediction lengths, suggesting that their predictive performance is independent of the prediction horizon. This observation indicates that RNN-based models may struggle to capture the inherent characteristics of time series data. Even in the case of AvgPred[1], there is a slight increase in MSE as the prediction horizon grows, although this change is relatively small.

---

[1]For the purpose of comparing experimental results, we introduce a simple model called **AvgPred**. **AvgPred** takes a multivariate time series as input and outputs a repetition of the average value of each input variable as the required length.

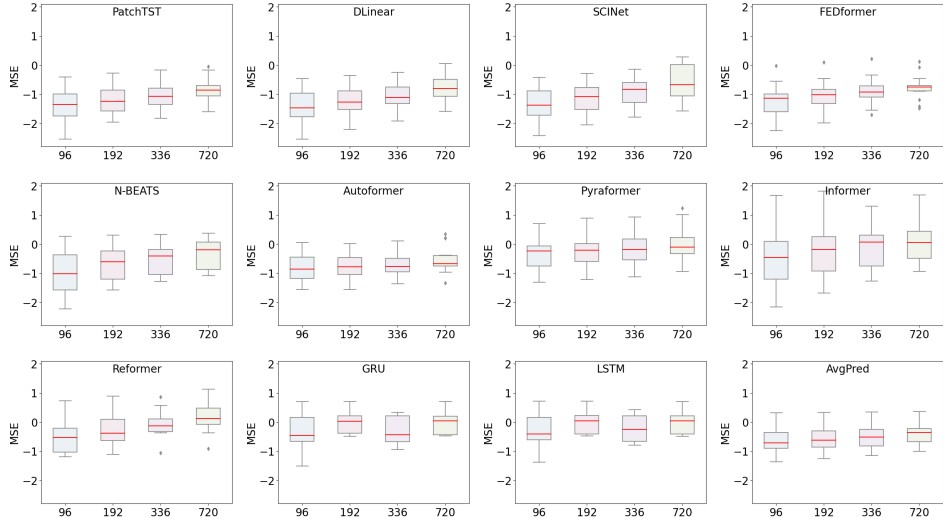

Figure 3: Box plot of MSE for the model at different prediction lengths. The horizontal red line represents the median. To enhance the clarity of the results in the graph, the MSE results have been log-transformed with base 8.

**Results of training epoch and time-consuming :** We conducted experiments to determine the number of epochs and training time required for model training. The results revealed that, regardless of the architecture on which the model is trained, in most cases, it only took **1 epoch** to achieve the lowest loss on the training dataset, and the results are very close to the model's published optimal performance, even achieve SOTA performance. This is probably because the existing single metric is insufficient to drive the model to achieve optimal performance. For specific experimental settings and detailed results, please refer to Appendix A.7.

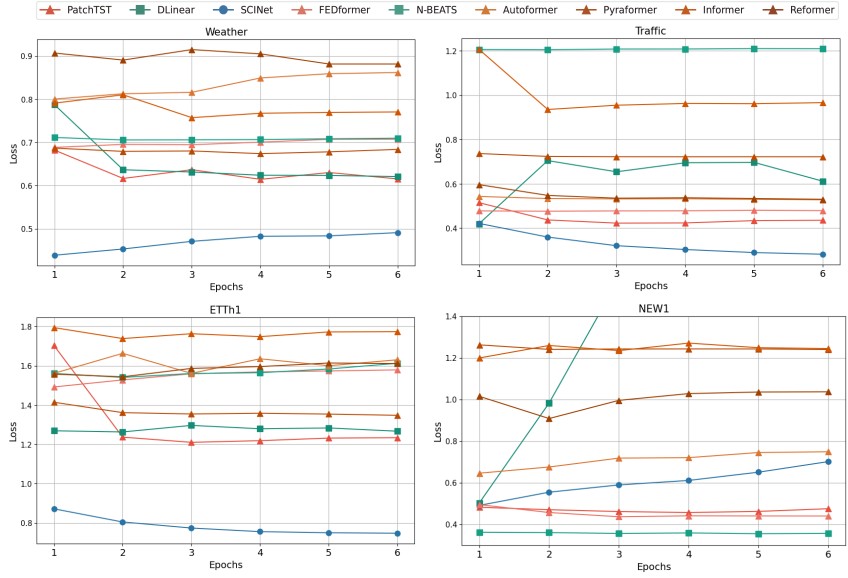

Figure 4: Model validation loss over traning epochs.

## 5 DEGENERACY

When studying the performance of different models on the LSTF problem, we observed that all models frequently exhibit degeneracy. **Degeneracy** refers to a phenomenon where the model's predictions maintain a low MSE value but consist of simple pattern repetition. Specifically, there are typically two scenarios:

The first type refers to the situation where the model only captures a certain level of periodicity but fails to learn other data characteristics, as shown in figure 5(a)(c)(d). In figure 5(a), even though PatchTST and DLinear models are based on different structures, they both capture similar periodicity and oscillation amplitudes. However, the models do not capture other characteristics of the data and consistently oscillate around the center line of the data. In figure 5(b)(c), the models also capture the periodicity of the data but are not as good as PatchTST and DLinear in controlling the oscillation amplitudes. Reformer and SCINet even learn opposite trends.

The second scenario is similar to the first one, but in this case, the model also fails to learn the periodicity of the data and instead predicts a straight line with a low slope, as illustrated in figure 5(b). The predictions made by N-BEATS completely degenerate into straight lines, while Informer exhibits a similar behavior. These predictions still have a low MSE because they remain close to the center line of the data. However, they fail to capture any meaningful data patterns. For predictions that completely degenerate into the mean of the input by using AvgPred, please refer to Appendix A.8.

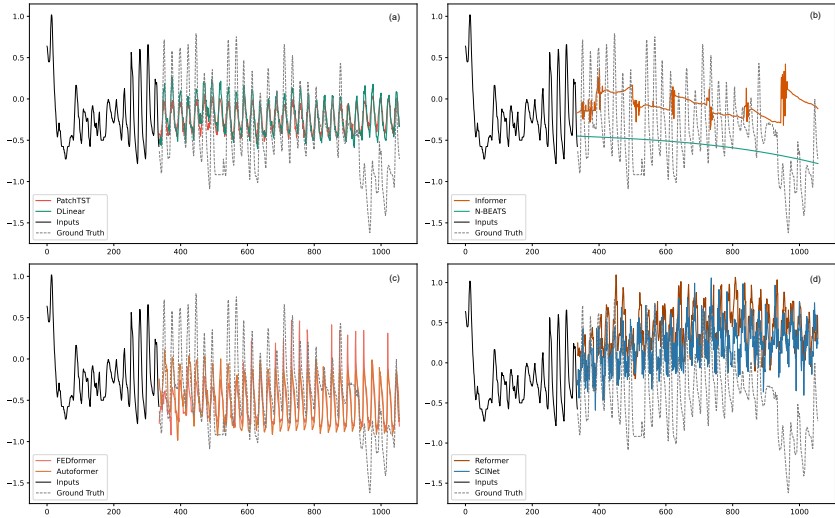

Figure 5: A sample illustrating model prediction degeneracy. The solid black line represents the model input data, the gray dashed line represents the ground truth, and the colored lines represent the model's predicted values.

## 6 CONCLUSION

In conclusion, we introduce 7 **NEW** time-series datasets and present a comprehensive evaluation of long sequence time-series forecasting models through the LST-Bench, addressing key aspects such as the overall prediction performance of models, and the models' generalization across different prediction lengths and datasets. We have observed a phenomenon named Degeneracy where the predictive performance deteriorates as models produce increasingly accurate MSE predictions. Additionally, we have noted that the model's optimal performance is achieved very early in the training process, often after just one epoch. This raises questions about the efficiency of training procedures and suggests potential opportunities for model optimization. These insights underscore the ongoing evolution and challenges in the field and pave the way for future research and innovations that will advance the effectiveness and applicability of time series forecasting methods.

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

# A APPENDIX

## A.1 FULL BENCHMARK RESULTS

Table 4: Sample table title

| Prediction | | sl336/pl96 | | sl336/pl192 | | sl336/pl336 | | sl336/pl720 | |
|---|---|---|---|---|---|---|---|---|---|
| | | MSE | MAE | MSE | MAE | MSE | MAE | MSE | MAE |
| Weather | PatchTST | 0.238 | 0.298 | **0.199** | **0.251** | **0.247** | **0.294** | **0.315** | 0.346 |
| | DLinear | 0.174 | 0.234 | 0.219 | 0.280 | 0.263 | 0.315 | 0.326 | 0.367 |
| | SCINet | **0.156** | **0.208** | 0.209 | 0.261 | 0.261 | 0.299 | 0.319 | **0.342** |
| | FEDformer | 0.250 | 0.337 | 0.285 | 0.355 | 0.359 | 0.399 | 0.404 | 0.415 |
| | N-BEATS | 0.158 | 0.221 | 0.206 | 0.274 | 0.275 | 0.330 | 0.350 | 0.383 |
| | Autoformer | 0.278 | 0.353 | 0.347 | 0.401 | 0.449 | 0.469 | 0.493 | 0.457 |
| | Pyraformer | 0.271 | 0.349 | 0.295 | 0.362 | 0.323 | 0.379 | 0.393 | 0.429 |
| | Informer | 0.243 | 0.312 | 0.316 | 0.368 | 0.433 | 0.450 | 0.442 | 0.460 |
| | Reformer | 0.360 | 0.401 | 0.507 | 0.513 | 0.743 | 0.626 | 0.790 | 0.662 |
| | GRU | OOM | OOM | OOM | OOM | OOM | OOM | OOM | OOM |
| | LSTM | OOM | OOM | OOM | OOM | OOM | OOM | OOM | OOM |
| Traffic | PatchTST | **0.429** | 0.305 | **0.440** | 0.304 | **0.457** | 0.316 | 0.494 | 0.329 |
| | DLinear | 0.430 | 0.316 | 0.443 | 0.319 | 0.459 | 0.331 | 0.488 | 0.345 |
| | SCINet | 0.442 | **0.281** | 0.454 | **0.283** | 0.470 | **0.291** | **0.484** | **0.303** |
| | FEDformer | 0.575 | 0.357 | 0.607 | 0.376 | 0.621 | 0.380 | 0.630 | 0.383 |
| | N-BEATS | 1.302 | 0.760 | 1.354 | 0.781 | 1.402 | 0.790 | 1.457 | 0.795 |
| | Autoformer | 0.644 | 0.415 | 0.643 | 0.401 | 0.620 | 0.385 | 0.677 | 0.418 |
| | Pyraformer | 0.908 | 0.502 | 0.903 | 0.495 | 0.913 | 0.497 | 0.936 | 0.506 |
| | Informer | 0.956 | 0.520 | 0.874 | 0.479 | 1.323 | 0.685 | 1.508 | 0.795 |
| | Reformer | 0.706 | 0.394 | 0.699 | 0.381 | 0.700 | 0.380 | 0.693 | 0.376 |
| | GRU | OOM | OOM | OOM | OOM | OOM | OOM | OOM | OOM |
| | LSTM | OOM | OOM | OOM | OOM | OOM | OOM | OOM | OOM |
| Electricity | PatchTST | **0.131** | **0.226** | **0.149** | **0.242** | 0.169 | 0.269 | **0.201** | **0.299** |
| | DLinear | 0.140 | 0.237 | 0.153 | 0.250 | **0.169** | **0.267** | 0.203 | 0.300 |
| | SCINet | 0.173 | 0.276 | 0.194 | 0.300 | 0.214 | 0.317 | 0.207 | 0.311 |
| | FEDformer | 0.188 | 0.304 | 0.196 | 0.311 | 0.212 | 0.327 | 0.244 | 0.352 |
| | N-BEATS | 0.760 | 0.715 | 0.806 | 0.740 | 0.841 | 0.759 | 0.872 | 0.772 |
| | Autoformer | 0.210 | 0.325 | 0.209 | 0.322 | 0.256 | 0.369 | 0.264 | 0.374 |
| | Pyraformer | 0.791 | 0.603 | 0.779 | 0.598 | 0.770 | 0.592 | 0.773 | 0.595 |
| | Informer | 0.322 | 0.409 | 0.420 | 0.476 | 0.499 | 0.517 | 0.953 | 0.785 |
| | Reformer | 0.302 | 0.403 | 0.328 | 0.421 | 0.345 | 0.431 | 0.399 | 0.475 |
| | GRU | OOM | OOM | OOM | OOM | OOM | OOM | OOM | OOM |
| | LSTM | OOM | OOM | OOM | OOM | OOM | OOM | OOM | OOM |
| ETTh1 | PatchTST | 0.388 | 0.412 | 0.429 | **0.436** | **0.455** | **0.459** | 0.500 | 0.507 |
| | DLinear | 0.387 | **0.408** | **0.427** | 0.439 | 0.470 | 0.468 | 0.503 | 0.513 |
| | SCINet | 0.402 | 0.425 | 0.467 | 0.469 | 0.543 | 0.521 | 0.858 | 0.695 |
| | FEDformer | **0.375** | 0.415 | 0.427 | 0.448 | 0.458 | 0.465 | **0.484** | **0.496** |
| | N-BEATS | 0.714 | 0.579 | 0.750 | 0.612 | 0.761 | 0.624 | 0.920 | 0.719 |
| | Autoformer | 0.531 | 0.501 | 0.546 | 0.508 | 0.505 | 0.505 | 0.526 | 0.530 |
| | Pyraformer | 1.115 | 0.828 | 1.140 | 0.846 | 1.241 | 0.903 | 1.268 | 0.924 |
| | Informer | 1.306 | 0.943 | 1.367 | 0.946 | 1.591 | 1.057 | 1.295 | 0.931 |
| | Reformer | 0.813 | 0.671 | 0.890 | 0.711 | 1.006 | 0.730 | 1.160 | 0.840 |
| | GRU | 0.776 | 0.635 | 1.090 | 0.804 | 1.244 | 0.847 | 1.133 | 0.808 |
| | LSTM | 0.897 | 0.698 | 1.128 | 0.811 | 1.215 | 0.837 | 1.171 | 0.825 |

Table 5: Sample table title

| Prediction | | sl336/pl96 MSE | MAE | sl336/pl192 MSE | MAE | sl336/pl336 MSE | MAE | sl336/pl720 MSE | MAE |
|---|---|---|---|---|---|---|---|---|---|
| ETTh2 | PatchTST | 0.320 | **0.380** | 0.404 | 0.434 | 0.438 | **0.460** | 0.710 | 0.598 |
| | DLinear | 0.369 | 0.416 | **0.375** | **0.410** | 0.481 | 0.479 | 0.844 | 0.658 |
| | SCINet | 0.416 | 0.446 | 0.617 | 0.551 | 0.744 | 0.607 | 1.314 | 0.827 |
| | FEDformer | 0.341 | 0.385 | 0.433 | 0.441 | 0.504 | 0.495 | **0.479** | **0.486** |
| | N-BEATS | 0.335 | 0.388 | 0.465 | 0.479 | 0.609 | 0.539 | 1.396 | 0.825 |
| | Autoformer | 0.447 | 0.487 | 0.469 | 0.504 | **0.430** | 0.468 | 0.668 | 0.634 |
| | Pyraformer | 2.009 | 1.129 | 2.429 | 1.275 | 2.537 | 1.311 | 2.754 | 1.414 |
| | Informer | 5.328 | 1.966 | 6.150 | 2.180 | 3.651 | 1.610 | 2.863 | 1.468 |
| | Reformer | 2.073 | 1.180 | 2.417 | 1.234 | 2.369 | 1.227 | 2.893 | 1.290 |
| | GRU | **0.280** | 0.392 | 0.742 | 0.735 | 0.561 | 0.625 | 0.622 | 0.649 |
| | LSTM | 0.343 | 0.449 | 0.654 | 0.674 | 0.547 | 0.616 | 0.616 | 0.633 |
| ETTm1 | PatchTST | 0.313 | 0.362 | 0.348 | 0.387 | 0.389 | 0.412 | 0.440 | 0.439 |
| | DLinear | **0.300** | **0.344** | **0.338** | **0.370** | **0.372** | **0.390** | **0.431** | **0.427** |
| | SCINet | 0.315 | 0.367 | 0.368 | 0.405 | 0.404 | 0.425 | 0.543 | 0.527 |
| | FEDformer | 0.363 | 0.413 | 0.394 | 0.425 | 0.440 | 0.456 | 0.499 | 0.484 |
| | N-BEATS | 0.420 | 0.431 | 0.676 | 0.545 | 0.719 | 0.568 | 0.777 | 0.611 |
| | Autoformer | 0.617 | 0.516 | 0.584 | 0.529 | 0.604 | 0.523 | 0.615 | 0.536 |
| | Pyraformer | 0.787 | 0.651 | 0.847 | 0.683 | 0.891 | 0.697 | 0.871 | 0.697 |
| | Informer | 0.576 | 0.558 | 0.670 | 0.647 | 0.897 | 0.738 | 0.895 | 0.714 |
| | Reformer | 0.760 | 0.639 | 0.966 | 0.711 | 0.984 | 0.718 | 1.105 | 0.779 |
| | GRU | 0.445 | 0.448 | 1.154 | 0.830 | 1.188 | 0.828 | 1.110 | 0.799 |
| | LSTM | 0.479 | 0.474 | 1.171 | 0.832 | 1.268 | 0.850 | 1.147 | 0.811 |
| ETTm2 | PatchTST | 0.170 | **0.256** | 0.227 | **0.299** | 0.301 | 0.347 | **0.408** | **0.416** |
| | DLinear | **0.168** | 0.257 | **0.224** | 0.303 | **0.278** | **0.336** | 0.654 | 0.562 |
| | SCINet | 0.192 | 0.289 | 0.309 | 0.380 | 0.498 | 0.495 | 1.169 | 0.771 |
| | FEDformer | 0.189 | 0.283 | 0.261 | 0.326 | 0.327 | 0.365 | 0.437 | 0.427 |
| | N-BEATS | 0.205 | 0.283 | 0.303 | 0.353 | 0.396 | 0.408 | 0.496 | 0.480 |
| | Autoformer | 0.336 | 0.396 | 0.361 | 0.411 | 0.406 | 0.433 | 0.467 | 0.451 |
| | Pyraformer | 0.805 | 0.678 | 1.016 | 0.800 | 1.258 | 0.938 | 3.458 | 1.636 |
| | Informer | 1.941 | 1.050 | 2.428 | 1.240 | 2.190 | 1.120 | 5.391 | 2.004 |
| | Reformer | 0.790 | 0.672 | 1.464 | 0.913 | 1.764 | 1.002 | 3.106 | 1.328 |
| | GRU | 0.220 | 0.350 | 0.777 | 0.744 | 0.562 | 0.585 | 0.667 | 0.661 |
| | LSTM | 0.252 | 0.387 | 0.785 | 0.755 | 0.625 | 0.645 | 0.708 | 0.691 |
| NEW1 | PatchTST | **0.079** | 0.163 | 0.141 | 0.251 | 0.161 | 0.261 | 0.318 | 0.425 |
| | DLinear | **0.079** | **0.161** | **0.111** | **0.200** | **0.146** | **0.237** | 0.247 | 0.344 |
| | SCINet | 0.088 | 0.169 | 0.128 | 0.213 | 0.167 | 0.252 | 0.242 | **0.312** |
| | FEDformer | 0.105 | 0.220 | 0.136 | 0.253 | 0.180 | 0.285 | **0.228** | 0.317 |
| | N-BEATS | 0.108 | 0.211 | 0.226 | 0.324 | 0.278 | 0.361 | 0.339 | 0.398 |
| | Autoformer | 0.252 | 0.349 | 0.264 | 0.362 | 0.355 | 0.420 | 0.380 | 0.427 |
| | Pyraformer | 0.505 | 0.559 | 0.574 | 0.599 | 0.606 | 0.602 | 0.697 | 0.650 |
| | Informer | 0.116 | 0.232 | 0.186 | 0.295 | 0.280 | 0.365 | 0.467 | 0.495 |
| | Reformer | 0.304 | 0.410 | 0.440 | 0.509 | 0.781 | 0.649 | 1.368 | 1.368 |
| | GRU | 0.599 | 0.635 | 0.612 | 0.645 | 0.390 | 0.476 | 0.625 | 0.654 |
| | LSTM | 0.664 | 0.671 | 0.677 | 0.680 | 0.474 | 0.521 | 0.667 | 0.675 |

Table 6: Sample table title

| Prediction | | sl336/pl96 | | sl336/pl192 | | sl336/pl336 | | sl336/pl720 | |
|---|---|---|---|---|---|---|---|---|---|
| | | MSE | MAE | MSE | MAE | MSE | MAE | MSE | MAE |
| NEW2 | PatchTST | 0.105 | 0.214 | 0.156 | 0.267 | 0.183 | 0.280 | 0.330 | **0.377** |
| | DLinear | **0.079** | **0.174** | **0.110** | **0.209** | **0.148** | **0.245** | 0.320 | 0.389 |
| | SCINet | 0.089 | 0.189 | 0.133 | 0.239 | 0.178 | 0.279 | 0.312 | 0.378 |
| | FEDformer | 0.143 | 0.274 | 0.183 | 0.308 | 0.233 | 0.343 | **0.304** | 0.387 |
| | N-BEATS | 0.118 | 0.234 | 0.231 | 0.341 | 0.292 | 0.382 | 0.390 | 0.432 |
| | Autoformer | 0.300 | 0.383 | 0.279 | 0.363 | 0.362 | 0.423 | 0.474 | 0.472 |
| | Pyraformer | 0.423 | 0.488 | 0.474 | 0.508 | 0.550 | 0.539 | 0.792 | 0.626 |
| | Informer | 0.134 | 0.239 | 0.243 | 0.322 | 0.327 | 0.359 | 0.392 | 0.395 |
| | Reformer | 0.309 | 0.403 | 0.515 | 0.531 | 0.778 | 0.645 | 1.680 | 1.060 |
| | GRU | 0.626 | 0.617 | 0.632 | 0.621 | 0.446 | 0.494 | 0.670 | 0.633 |
| | LSTM | 0.619 | 0.609 | 0.623 | 0.611 | 0.455 | 0.489 | 0.658 | 0.625 |
| NEW3 | PatchTST | 0.210 | 0.321 | 0.269 | 0.372 | **0.312** | **0.402** | **0.400** | **0.463** |
| | DLinear | **0.200** | **0.311** | **0.265** | **0.366** | 0.323 | 0.412 | 0.455 | 0.502 |
| | SCINet | 0.233 | 0.344 | 0.361 | 0.436 | 0.554 | 0.545 | 0.825 | 0.676 |
| | FEDformer | 0.304 | 0.409 | 0.346 | 0.435 | 0.383 | 0.461 | 0.465 | 0.510 |
| | N-BEATS | 0.255 | 0.359 | 0.321 | 0.412 | 0.361 | 0.437 | 0.431 | 0.485 |
| | Autoformer | 0.379 | 0.460 | 0.451 | 0.506 | 0.478 | 0.507 | 0.483 | 0.509 |
| | Pyraformer | 0.459 | 0.517 | 0.559 | 0.573 | 0.698 | 0.649 | 0.974 | 0.777 |
| | Informer | 0.290 | 0.395 | 0.386 | 0.467 | 0.461 | 0.510 | 0.556 | 0.562 |
| | Reformer | 0.495 | 0.537 | 0.663 | 0.639 | 1.018 | 0.781 | 1.110 | 0.820 |
| | GRU | 0.615 | 0.610 | 0.619 | 0.613 | 0.462 | 0.505 | 0.627 | 0.626 |
| | LSTM | 0.631 | 0.615 | 0.634 | 0.617 | 0.490 | 0.527 | 0.630 | 0.624 |
| NEW4 | PatchTST | 0.506 | 0.404 | 0.530 | 0.435 | **0.621** | **0.478** | **0.844** | **0.571** |
| | DLinear | **0.397** | **0.373** | **0.493** | **0.422** | 0.626 | 0.487 | 0.905 | 0.616 |
| | SCINet | 0.514 | 0.415 | 0.636 | 0.476 | 0.785 | 0.545 | 1.072 | 0.677 |
| | FEDformer | 0.543 | 0.458 | 0.628 | 0.495 | 0.713 | 0.523 | 0.928 | 0.605 |
| | N-BEATS | 0.625 | 0.466 | 0.803 | 0.561 | 0.998 | 0.631 | 1.202 | 0.710 |
| | Autoformer | 0.900 | 0.624 | 1.017 | 0.667 | 1.096 | 0.676 | 1.419 | 0.760 |
| | Pyraformer | 1.084 | 0.674 | 1.246 | 0.726 | 1.385 | 0.767 | 1.533 | 0.833 |
| | Informer | 1.150 | 0.676 | 1.302 | 0.740 | 1.333 | 0.732 | 1.660 | 0.821 |
| | Reformer | 0.979 | 0.620 | 1.228 | 0.729 | 1.474 | 0.807 | 1.687 | 0.867 |
| | GRU | 2.031 | 0.964 | 2.029 | 0.961 | 1.387 | 0.798 | 2.012 | 0.954 |
| | LSTM | 2.040 | 0.970 | 2.036 | 0.965 | 1.539 | 0.847 | 2.017 | 0.956 |
| NEW5 | PatchTST | 0.191 | 0.296 | 0.240 | 0.342 | 0.297 | 0.385 | 0.409 | 0.461 |
| | DLinear | **0.181** | **0.283** | **0.220** | **0.320** | **0.276** | **0.369** | **0.395** | **0.455** |
| | SCINet | 0.194 | 0.303 | 0.251 | 0.354 | 0.327 | 0.413 | 0.435 | 0.485 |
| | FEDformer | 0.239 | 0.353 | 0.290 | 0.393 | 0.350 | 0.432 | 0.444 | 0.485 |
| | N-BEATS | 0.393 | 0.391 | 0.639 | 0.524 | 0.802 | 0.598 | 0.986 | 0.671 |
| | Autoformer | 0.876 | 0.650 | 0.860 | 0.637 | 1.017 | 0.694 | 1.220 | 0.773 |
| | Pyraformer | 0.450 | 0.494 | 0.488 | 0.518 | 0.534 | 0.544 | 0.671 | 0.631 |
| | Informer | 0.690 | 0.549 | 0.956 | 0.670 | 1.273 | 0.792 | 1.156 | 0.759 |
| | Reformer | 0.355 | 0.447 | 0.650 | 0.641 | 0.709 | 0.660 | 0.945 | 0.781 |
| | GRU | 1.323 | 0.857 | 1.324 | 0.858 | 1.242 | 0.793 | 1.317 | 0.858 |
| | LSTM | 1.324 | 0.857 | 1.326 | 0.859 | 1.201 | 0.780 | 1.317 | 0.858 |

Table 7: Sample table title

| Prediction | | sl336/pl96 | | sl336/pl192 | | sl336/pl336 | | sl336/pl720 | |
|---|---|---|---|---|---|---|---|---|---|
| | | MSE | MAE | MSE | MAE | MSE | MAE | MSE | MAE |
| NEW6 | PatchTST | 0.284 | 0.364 | 0.306 | 0.392 | 0.377 | 0.440 | 0.434 | 0.491 |
| | DLinear | 0.263 | 0.356 | **0.294** | **0.385** | **0.332** | **0.419** | 0.432 | 0.497 |
| | SCINet | 0.275 | 0.366 | 0.316 | 0.407 | 0.357 | 0.435 | 0.469 | 0.512 |
| | FEDformer | 0.347 | 0.430 | 0.377 | 0.452 | 0.407 | 0.473 | 0.497 | 0.529 |
| | N-BEATS | **0.207** | **0.322** | 0.296 | 0.396 | 0.347 | 0.437 | **0.414** | **0.484** |
| | Autoformer | 0.405 | 0.488 | 0.413 | 0.488 | 0.380 | 0.468 | 0.427 | 0.490 |
| | Pyraformer | 0.530 | 0.552 | 0.554 | 0.569 | 0.570 | 0.576 | 0.614 | 0.606 |
| | Informer | 0.471 | 0.529 | 0.795 | 0.670 | 0.652 | 0.648 | 0.810 | 0.713 |
| | Reformer | 0.454 | 0.501 | 0.561 | 0.576 | 0.689 | 0.649 | 0.923 | 0.763 |
| | GRU | 1.032 | 0.836 | 1.034 | 0.834 | 0.648 | 0.643 | 1.037 | 0.833 |
| | LSTM | 1.038 | 0.838 | 1.044 | 0.838 | 0.776 | 0.708 | 1.042 | 0.836 |
| NEW7 | PatchTST | 0.663 | 0.504 | 0.762 | 0.552 | 0.837 | 0.594 | **0.944** | **0.654** |
| | DLinear | **0.629** | **0.487** | **0.699** | **0.531** | **0.782** | **0.574** | 1.050 | 0.700 |
| | SCINet | 0.654 | 0.527 | 0.744 | 0.566 | 0.861 | 0.619 | 1.099 | 0.720 |
| | FEDformer | 0.979 | 0.643 | 1.094 | 0.667 | 1.241 | 0.733 | 1.123 | 0.734 |
| | N-BEATS | 0.754 | 0.588 | 0.945 | 0.668 | 1.027 | 0.712 | 1.096 | 0.768 |
| | Autoformer | 1.060 | 0.744 | 1.000 | 0.725 | 1.108 | 0.775 | 1.233 | 0.819 |
| | Pyraformer | 0.943 | 0.702 | 0.992 | 0.725 | 1.070 | 0.761 | 1.215 | 0.838 |
| | Informer | 0.882 | 0.702 | 1.217 | 0.861 | 1.366 | 0.926 | 1.563 | 0.999 |
| | Reformer | 1.177 | 0.768 | 1.136 | 0.793 | 1.141 | 0.805 | 1.463 | 0.954 |
| | GRU | 2.001 | 1.120 | 1.996 | 1.119 | 1.333 | 0.916 | 1.978 | 1.112 |
| | LSTM | 2.026 | 1.142 | 2.020 | 1.140 | 1.439 | 0.959 | 1.997 | 1.130 |

## A.2 MODEL IMPLEMENT

**Informer** was acquired at https://github.com/zhouhaoyi/Informer2020. During the experiment, the hyperparameter label-len was set to 168, half of seq-len, and other hyperparameters were set to default settings.

**Autoformer** and **Reformer** was acquired at https://github.com/thuml/Autoformer. The hyperparameter setting method is the same as Informer.

**FEDformer** was acquired at https://github.com/MAZiqing/FEDformer. During the experiment, hyperparameters were set to default settings.

**N-BEATS** was acquired at https://github.com/philipperemy/n-beats. During the experiment, certain modifications were made to it so that it can perform multi-variable predictions: we exchange the last two dimensions of the input data with a shape of [batchSize, seqLen, features] during training. But we did not modify the hyperparameters.

**PatchTST** was acquired at https://github.com/yuqinie98/PatchTST. During the experiment, hyperparameters were set to default settings.

**DLinear** We used the implementation of DLinear from the PatchTST GitHub repository. During the experiment, hyperparameters were set to default settings.

**SCINet** was acquired at https://github.com/cure-lab/SCINet. During the experiment, hyperparameters were set to default settings.

**Pyraformer** was acquired at https://github.com/ant-research/Pyraformer. During the experiment, hyperparameters were set to default settings.

## A.3 Models' performance across different prediction length

We assume that, by using the same model, the prediction results should be more and more inaccurate with the prediction length becomes longer. We initially conducted a comparative analysis of the models' predictive performance at prediction horizons of 96 and 720, as outlined in Table 8. It is noteworthy that as the prediction horizon extended to 720, all models exhibited a decline in performance; however, the extent of this decline varied across models. Among the 15 datasets considered, the most significant percentage decrease in predictive performance was observed in the case of the Reformer model, with a reduction of up to 199.9%. In contrast, the GRU model exhibited the least decline, at a modest 27.1%. It is important to highlight that the GRU model does not attain state-of-the-art predictive performance by itself. Among the models belonging to the top tier in terms of predictive accuracy, both PatchTST and DLinear showed relatively moderate declines, with respectively reductions of 36.4% and 36.5%. In contrast, the performance of SCINet experienced a substantial decrease, amounting to 79.7%.

Table 8: Model performance decline percentage

| Model | PatchTST | DLinear | SCINet | FEDformer | N-BEATS | Autoformer |
|---|---|---|---|---|---|---|
| performance decline 100% | 0.419 | 0.473 | 0.514 | 0.341 | 0.464 | 0.218 |
| Model | Pyraformer | Informer | Reformer | GRU | LSTM | AvgPred |
| performance decline 100% | 0.273 | 0.361 | 0.450 | 0.204 | 0.176 | 0.193 |

## A.4 Results of training time consuming

To compare the training time of the models, we conducted experiments with a training batch size of 32, and an early stopping tolerance of 5 for each model. The maximum training epochs for each model were set to 50.

## A.5 Stability of models' ranking

It is noteworthy that, when comparing the results in PatchTST, where the authors re-ran FEDformer, Autoformer, and Informer using six different look-back windows and selected the best-performing configuration, the rankings of model performance remained consistent. This observation underscores the presence of a performance barrier among the four tiers of models. Although there is indeed variability in model performance, exemplified by Pyraformer's mean squared error values on the Weather and Electricity datasets for prediction lengths in $\{96, 192, 336, 720\}$, which were reported as $\{0.896, 0.622, 0.739, 1.004\}$ and $\{0.386, 0.386, 0.378, 0.376\}$ in PatchTST, and observed as $\{0.280, 0.288, 0.329, 0.364\}$ and $\{0.791, 0.779, 0.770, 0.773\}$ in our own results, this variability does not significantly impact the model's ranking on individual datasets, nor does it influence the overall ranking across all datasets.

## A.6 Details of datasets

**ETT** (Electricity Transformer Temperature)[2]: In the electric power industry, the ETT is widely used as an indicator in equipment monitoring. The dataset of ETT contains 2-year ETT data from two separated countries in China. We create separate datasets as {ETTh1, ETTh2} for 1-hour level and {ETTm1, ETTm2} for 15-minute-level. Each data point consists of the target value "oil temperature" and 6 power load features.

**ECL** (Electricity Consuming Load)[3]: The ECL dataset collects the electricity consumption (Kwh) of 321 clients, which is powerful in the analysis of the behavior in electricity consumption. We set 'MT 320' as the target value.

---

[2]ETT dataset was acquired at https://github.com/zhouhaoyi/ETDataset.
[3]ECL dataset was acquired at https://archive.ics.uci.edu/ml/ datasets/ElectricityLoadDiagrams20112014.

**Weather**[4]: The Weather dataset contains local climatological data from 21 meteorological indicators in Germany. The data points are collected every 1 hour.

**Traffic**[5]: The traffic dataset is a collection of hourly data from California Department of Transportation, which describes the road occupancy rates (between 0 and 1) measured by different sensors on San Francisco Bay area freeways.

## A.7 Training Epochs and Time-consuming

Table 9: Model performance on training epochs and time-consuming

| Model | Dataset | Training Epochs | Total Time | Epoch Time |
|---|---|---|---|---|
| Reformer | ETTh1 | 7 | 445.355 | 63.622 |
| | NEW1 | 7 | 1874.410 | 267.773 |
| | Weather | 10 | 2873.441 | 287.344 |
| | Traffic | 22 | 3398.867 | 154.494 |
| DLinear | ETTh1 | 19 | 76.129 | 4.007 |
| | NEW1 | 6 | 100.824 | 16.804 |
| | Weather | 15 | 385.276 | 25.685 |
| | Traffic | 6 | 909.391 | 151.565 |
| FEDformer | ETTh1 | 6 | 885.595 | 147.599 |
| | NEW1 | 8 | 5116.847 | 639.606 |
| | Weather | 6 | 4109.424 | 684.904 |
| | Traffic | 7 | 2320.914 | 331.559 |
| Informer | ETTh1 | 7 | 265.129 | 37.876 |
| | NEW1 | 6 | 1002.969 | 167.162 |
| | Weather | 8 | 1445.666 | 180.708 |
| | Traffic | 7 | 1093.404 | 156.201 |
| Autoformer | ETTh1 | 8 | 415.201 | 51.900 |
| | NEW1 | 6 | 1349.438 | 224.906 |
| | Weather | 6 | 1438.323 | 239.721 |
| | Traffic | 11 | 1648.682 | 149.880 |
| Pyraformer | ETTh1 | 22 | 743.286 | 33.786 |
| | NEW1 | 10 | 766.974 | 76.697 |
| | Weather | 13 | 1245.314 | 95.793 |
| | Traffic | 13 | 1961.292 | 150.869 |
| PatchTST | ETTh1 | 12 | 124.752 | 10.396 |
| | NEW1 | 9 | 445.306 | 49.479 |
| | Weather | 14 | 684.931 | 48.924 |
| | Traffic | 8 | 3173.363 | 396.670 |
| SCINet | ETTh1 | 12 | 171.018 | 14.252 |
| | NEW1 | 6 | 1035.571 | 172.595 |
| | Weather | 6 | 423.238 | 70.540 |
| | Traffic | 21 | 7217.767 | 343.703 |
| N-BEATS | ETTh1 | 7 | 23.986 | 3.427 |
| | NEW1 | 11 | 251.665 | 22.879 |
| | Weather | 8 | 248.152 | 31.019 |
| | Traffic | 10 | 6699.644 | 669.964 |

## A.8 AvgPred performance

---

[4]Weather dataset was acquired at https://www.bgc-jena.mpg.de/wetter/
[5]Traffic dataset was acquired at http://pems.dot.ca.gov.

Table 10: Average Prediction Performance

| Prediction | sl336/pl96 | | sl336/pl192 | | sl336/pl336 | | sl336/pl720 | |
|---|---|---|---|---|---|---|---|---|
| | MSE | MAE | MSE | MAE | MSE | MAE | MSE | MAE |
| Weather | 0.256 | 0.311 | 0.284 | 0.328 | 0.317 | 0.347 | 0.368 | 0.380 |
| Traffic | 1.378 | 0.794 | 1.394 | 0.798 | 1.409 | 0.801 | 1.429 | 0.804 |
| Electricity | 0.834 | 0.757 | 0.844 | 0.760 | 0.857 | 0.764 | 0.883 | 0.773 |
| ETTh1 | 0.706 | 0.567 | 0.713 | 0.575 | 0.706 | 0.579 | 0.700 | 0.596 |
| ETTh2 | 0.385 | 0.419 | 0.401 | 0.430 | 0.393 | 0.428 | 0.432 | 0.455 |
| ETTm1 | 0.680 | 0.544 | 0.689 | 0.549 | 0.703 | 0.558 | 0.721 | 0.570 |
| ETTm2 | 0.490 | 0.481 | 0.539 | 0.504 | 0.593 | 0.530 | 0.718 | 0.588 |
| NEW1 | 0.337 | 0.389 | 0.357 | 0.399 | 0.386 | 0.413 | 0.442 | 0.434 |
| NEW2 | 0.448 | 0.494 | 0.471 | 0.506 | 0.511 | 0.526 | 0.602 | 0.567 |
| NEW3 | 0.350 | 0.438 | 0.379 | 0.457 | 0.412 | 0.474 | 0.489 | 0.518 |
| NEW4 | 0.700 | 0.570 | 0.771 | 0.594 | 0.863 | 0.626 | 1.064 | 0.690 |
| NEW5 | 0.473 | 0.519 | 0.501 | 0.536 | 0.543 | 0.560 | 0.630 | 0.606 |
| NEW6 | 0.427 | 0.473 | 0.445 | 0.488 | 0.473 | 0.509 | 0.527 | 0.549 |
| NEW7 | 0.812 | 0.612 | 0.867 | 0.633 | 0.945 | 0.662 | 1.111 | 0.724 |

