# OpenReview forum: "LST-Bench:A Benchmark for long sequence time-series forecasting Task"
_ICLR.cc/2024/Conference — Submitted to ICLR 2024_

### Official Review · Reviewer_inoa · 2023-10-27

**Soundness:** 2 fair
**Presentation:** 1 poor
**Contribution:** 1 poor
**Rating:** 3
**Confidence:** 4

**Summary:**

In spite of the widespread use of time series forecasting models, there are limited benchmarks for assessing the quality of different models in specific settings. In this work, the authors focus on model performance when applied to long-horizon forecasting, and demonstrate that all models perform poorly for datapoints further into the future - the authors refer to this problem as “degeneracy”. Although some existing models perform better than others, the analysis reveals a wider problem, and the authors call for further research into why performance degrades in such settings.

**Strengths:**

- Identifying limitations in existing models is helpful for informing future research directions, and addressing potential misconceptions on expected performance.
- The authors present a clear comparison of how different models compare against a suite of datasets for long-horizon predictions. The obtained rankings could be helpful for practitioners assessing which model to apply to their problem setting.

**Weaknesses:**

- To my understanding, the 9 new datasets all originate from the same problem domain (equipment monitoring). The authors did not share the time series yet; however, I would not expect there to be much diversity among these new time series. A well-developed benchmark should cover additional time series from different domains, and hence I don’t see this addition as being particularly noteworthy.
- The insights gleaned from the paper are very limited, as the authors primarily present empirical evidence of the degraded performance over longer horizons without much novel perceptions. Although I appreciate the evaluation presented by the authors in this work, I don’t think the paper feels complete without a clearer vision of how the issues identified here can be effectively resolved or at least partially mitigated in a tangible manner.
- In the evaluation, the authors comment on how the default hyper-parameters were selected for the models under review. I can see how this simplifies the experimental set-up, but I’m also left wondering about the extent to which adequate hyper-parameter optimisation could improve the reported results.
- Unfortunately, the paper’s writing isn’t very compelling, and the overall structuring seems more akin to a technical report than a conference paper.

**Questions:**

I list out my concerns around the paper in the *Weaknesses* section. I would be looking for responses to those comments in the forthcoming author rebuttal.

---

### Official Review · Reviewer_PPjW · 2023-10-29

**Soundness:** 2 fair
**Presentation:** 3 good
**Contribution:** 2 fair
**Rating:** 3
**Confidence:** 4

**Summary:**

This paper presents a long sequence timeseries benchmark framework that involves 11 benchmark models and 14 datasets, including 7 new and 7 old. Specifically, the authors conducted a thorough analysis on the experimental results, including the overall prediction performance of models and their generalization across different prediction lengths and datasets. The authors found a phenomenon referred to as "degeneracy", which means the low MSE value but with predictions having repetitive and simplistic pattern generations. Additionally, the authors found that the optimal performance is very close to its performance after training just one epoch. The authors claimed their benchmark work serves as valuable resource for advancing research in the field of timeseries forecasting.

**Strengths:**

I think this paper is easy to follow and presents many detailed results on the long sequence timeseries benchmark model performance, covering different domains. Also the new datasets may be used in other works for model evaluation. The experimental setup and configuration detail is very clear.

**Weaknesses:**

1. The novelty in this paper is very marginal. Though the authors have presented many results, the originality of the paper is very low. Nothing is new in terms of technical frameworks. All models are existing and some datasets have been public. Though the new datasets are provided for evaluating the models, it is unclear to me how much value it would bring for the community.

2. The technical detail for each model is missing. The authors should present each model architecture in the paper for completeness at least.

3. The authors have found some interesting phenomena, while they failed to investigate them to dig out the underlying cause. They have been quite generically existing in different scenarios. If the authors could study them in a theoretical perspective, this would add great value to the paper.

4. More accuracy metrics should be introduced in the paper to extend the generality of the benchmark framework, such as MAPE, CVRMSE, NMBE.

**Questions:**

1. How to improve the paper's novelty?
2. What is the underlying theoretical cause of the "degeneracy" in the work?

---

### Official Review · Reviewer_t1kK · 2023-10-29

**Soundness:** 2 fair
**Presentation:** 2 fair
**Contribution:** 2 fair
**Rating:** 3
**Confidence:** 4

**Summary:**

This paper provides a benchmark for long sequence time series forecasting (LSTF) task. The benchmark consists of 4 ETT datasets, weather, traffic, electricity and 7 NEW datasets of 15-minute-level data from the power industry. By conducting experiments of 11 major forecasting models on these datasets, the paper reveals the phenomenon of Degeneracy, that is the model's predictions consistently maintain a low MSE loss but are characterized by repetitive and simplistic pattern generation. Meanwhile, these model achieve near convergence after just one epoch. These two features need further investigation.

**Strengths:**

1. This paper can be considered as the first benchmark for LSTF problem after the widespread application of deep learning in time series forecasting, and serve as a valuable resource for future research.
2. This paper reveals two important phenomenon: Degeneracy and quick-convergence in LSTF problem, which inspires future investigation.

**Weaknesses:**

The paper summarizes the experiment results, but lacks further discussion about those phenomenon. For example, will Degeneracy affect generalization ability of the model? Any showcase illustrates the differences between one-epoch model and fully-trained model? I think by diving into those questions will make the paper more inspiring.

**Questions:**

see weakness

---

### Official Review · Reviewer_1Wsy · 2023-10-31

**Soundness:** 1 poor
**Presentation:** 2 fair
**Contribution:** 1 poor
**Rating:** 1
**Confidence:** 5

**Summary:**

This paper presents LST-Bench to evaluate long-sequence time-series forecasting models. The benchmark includes four types of models and evaluates them on a variety of datasets, including a new dataset called NEW. The experimental results show that all models exhibit significantly faster convergence compared to models in other domains, but also exhibit a phenomenon called Degeneracy, which signifies that the model's predictions exhibit lower MSE/MAE values, yet in fact, the forecasted outcomes tend to manifest as periodic repetitions, failing to genuinely predict the future data. The paper emphasizes the need for further investigation to address this issue and to reevaluate or redefine evaluation metrics applicable to time series forecasting to more accurately measure model performance, model design, and improvement on that. The paper also discusses the limitations of the benchmark, such as the input being limited to a single time series and not allowing for the inclusion of external features.

**Strengths:**

The authors conduct a series of experiments on existing models and datasets and spot the phenomenon called Degeneracy.

The authors collect a dataset from the electricity industry.

**Weaknesses:**

1. The contribution and novelty of this paper are low. The authors have applied existing time series models to commonly used datasets, which does not provide additional insights into either the model or its results.

2. The only new thing in this work seems to be the introduction of a NEW dataset from the electricity industry. Unfortunately, the authors have not provided any additional information about this dataset beyond its source and sampling rate.

**Questions:**

NA

---

### Meta-Review · Area_Chair_D5qA · 2023-12-05

**Metareview:**

This paper introduces a benchmark for long-term time-series forecasting, comparing 11 major models on 14 datasets. While such a benchmark would be useful if it is designed well, there are concerns about the novelty and contribution of the paper. The phenomena observed from the experiments could potentially bring about insights that can lead to further research. Unfortunately, those phenomena were not discussed and analyzed in detail in the paper. The authors are encouraged to consider the comments and suggestions of the reviewers to improve the paper.

**Justification For Why Not Higher Score:**

It is well below the acceptance standard.

**Justification For Why Not Lower Score:**

N/A

---

### Decision · Program_Chairs · 2024-01-16

Reject